# Treating Cognition in Schizophrenia: A Whole Lifespan Perspective

**DOI:** 10.3390/healthcare12212196

**Published:** 2024-11-04

**Authors:** Rafael Penadés, Maria Florencia Forte, Gisela Mezquida, Claudia Andrés, Rosa Catalán, Bàrbara Segura

**Affiliations:** 1Hospital Clínic of Barcelona, University of Barcelona, IDIBAPS, CIBERSAM, 08036 Barcelona, Spain; forte@recerca.clinic.cat (M.F.F.); candresg@clinic.cat (C.A.); rcatalan@clinic.cat (R.C.); 2Serra-Hunter Lecturer Fellow, University of Barcelona, IDIBAPS, CIBERSAM, 08036 Barcelona, Spain; mezquida@recerca.clinic.cat; 3Institute of Neurosciences, University of Barcelona, IDIBAPS, 08036 Barcelona, Spain; bsegura@ub.edu

**Keywords:** cognition, cognitive remediation, early psychosis, elderly, schizophrenia

## Abstract

**Background/Objectives:** Cognitive impairment is a core feature of schizophrenia, affecting attention, memory, and executive function and contributing significantly to the burden of the disorder. These deficits often begin before the onset of psychotic symptoms and persist throughout life, making their treatment essential for improving outcomes and functionality. This work aims to explore the impact of these impairments at different life stages and the interventions that have been developed to mitigate their effects. **Methods:** This narrative review examined literature searching for different approaches to treat cognitive impairments in schizophrenia across the lifespan. **Results:** Cognitive alterations appear before psychosis onset, suggesting a window for primary prevention. Then, a period of relative stability with a slight decline gives the period to secondary and eventually tertiary prevention for more than two decades. Finally, another window for tertiary prevention occurs from the third decade of illness until the later stages of the illness, when a progression in cognitive decline could be accelerated in some cases. Cognitive remediation and physical exercise are evidence-based interventions that should be provided to all patients with disabilities. **Conclusions:** Treating cognition throughout the whole lifespan is crucial for improving functional outcomes. It is necessary to consider the need for personalized, stage-specific strategies to enhance cognitive function and functioning in patients.

## 1. Introduction

Cognitive impairment is a core feature of schizophrenia that significantly contributes to the overall burden of the disease [1]. Cognition, encompassing a range of mental processes such as reasoning, learning, and problem-solving, is a critical component of daily functioning. It can affect attention, memory, executive function, and processing speed in schizophrenia, being one of the most important factors causing social disability [2]. Cognitive impairment often emerges before the onset of psychotic symptoms and tends to persist throughout the lifespan [3]. Addressing these impairments is crucial for improving the functional outcomes and quality of life for patients with schizophrenia.

From a neuroanatomical and physiological perspective, cognitive impairment seems to be caused by different brain changes in brain structure [1]. The most reported findings are cortical thickness, ventriculomegaly, and loss of dendritic spines in pyramidal neurons of the prefrontal cortex [4]. Those brain changes appear to have taken place in the early stages of neurodevelopment. Still, it is presumed that a cumulative effect of neurodevelopmental abnormalities is lifelong, producing enduring alterations in neuroplasticity and neuronal activity [5].

The course of cognition across the lifespan and its impact on functionality in persons with schizophrenia is now an open question. Different trajectories in cognitive functioning have been proposed: stable impairment, progressive decline, improvement, or fluctuating course [6]. Understanding cognitive trajectories, from the early phases of psychosis to the chronic stages of schizophrenia, could lead to more personalized and effective treatments [7].

This narrative review focuses on the treatment of cognitive impairments in schizophrenia across the lifespan. It aims to explore the mechanisms underlying cognitive deficits in schizophrenia, the impact of these impairments at different life stages, and the interventions that have been developed to mitigate their effects. By integrating research from neuropsychology, psychiatry, pharmacology, and cognitive rehabilitation, this paper will overview current strategies and emerging approaches to treating cognition in patients with schizophrenia.

## 2. Materials and Methods

This is a subjective, selective rather than a systematic review. As it is a theoretical integrative review, we aimed to analyze the available theories addressing the treatment of cognition throughout the whole lifespan in patients with schizophrenia based on cognitive trajectory analysis and available evidence-based treatments. The literature search was conducted on 27 May 2024 on two electronic databases (PubMed and Scopus). Abstracts were searched from 2000 to 2024. Search terms were divided into two different parts. The first is about cognitive trajectory, and the other is about cognitive treatments. Regarding cognitive trajectory, search terms were (‘psychosis’ OR ‘schizophrenia’ OR ‘schizo*’ OR ‘FEP’) OR (‘first episode psychosis’) AND (‘cog*’, ‘neurops*’ OR ‘neurocognit*’ OR ‘memory’ OR ‘attention’, OR ‘executive’ or ‘processing’) AND (‘longitudinal’ OR ‘follow-up’ OR ‘chang*’ OR ‘course’ OR ‘trajectory’). In addition, search terms for treatments were (Schizophrenia OR “psychosis” OR “psychotic”) AND ((cognit* OR “processing” OR “attention” OR “memory” OR “executive”) AND (“remediation” OR “rehabilitation” OR “enhancement” OR “training” OR “treatment” OR “therapy” OR antipsychotic* OR “molecule” OR “stimulation” OR “technique” OR “intervention” OR “exercise”)) AND (“meta-analysis” OR “systematic review”). Article titles and abstracts were reviewed to identify relevance, with full texts then being examined. Publications were selected based on their representativeness of the topic, prioritizing those with the best methodological quality and greatest representativeness as it is a narrative review. The meta-analysis, systematic reviews, controlled trials, and international clinical guidelines were always preferred.

## 3. Results

A total of 522 publications were first identified via abstracts and titles in databases (n = 508) and other sources (n = 14). Literature on cognitive trajectories and cognitive treatments in schizophrenia, while important, is not as abundant as some other areas of research. The number of studies specifically addressing these topics across the entire lifespan of individuals with schizophrenia is somewhat limited. Results reveal a significant number of studies addressing these topics, reflecting extensive research interest in understanding the progression of cognitive impairments in schizophrenia and the effectiveness of various treatments. The literature covers a wide range of aspects, including early intervention as primary prevention, age-related cognitive changes and progression, and the efficacy of cognitive remediation and pharmacological interventions in later life stages (Figure 1). 

### 3.1. Cognitive Trajectories

The cognitive impairment described in schizophrenia is not the same throughout the life cycle, showing different degrees of impairment at various stages. However, there is no agreement on the description of evolution and the following deterioration. Two alternative views have been proposed, some postulating a progressive and continuous deterioration while others advocate for a course in which deterioration appears in the early stages and remains relatively stable during the rest of the life cycle. However, different studies have provided evidence of both models, which may lead us to adopt an integrative model.

The most established theories have proposed that cognitive impairments appear before the onset of symptoms of the disease, and from that moment on, they remain stable throughout life [8]. In addition, different longitudinal studies have confirmed that cognitive impairment remains stable at least during the first 5 years of the disease [9]. However, there is probably no single pattern for all affected individuals. For example, patients who experience psychotic symptoms that do not remit with antipsychotic treatment often experience progressive cognitive decline [10].

When follow-up studies are conducted over long periods, we find results that suggest some decline. Ten-year follow-up studies demonstrated a significantly greater progressive cognitive decline than in healthy controls [11]. These results have been confirmed in studies with longer follow-ups of 20 years [12,13]. In summary, the evidence is mixed but suggests a slow cognitive decline is observed after the onset of the disease.

When investigating cognitive function in later life in older adults with chronic schizophrenia, the results are similar. A meta-analysis [14] showed widespread impairments in immediate memory and executive functioning. However, longitudinal studies by the same authors failed to show a pronounced decline [14]. In all, these findings suggest that while cognitive changes in chronic schizophrenia resemble typical age-related decline, they are likely to be more pronounced. Further research suggests how those deficits may be compounded by an age-related cognitive and functional decline [15,16]. However, other studies [17] propose that this cognitive decline may not only be more pronounced but may also occur at an earlier age as a form of “premature cognitive aging”.

Considering the data reviewed above, we could reach the following tentative hypothesis: cognitive decline would begin before psychosis onset, suggesting a window for primary prevention. Then, a period of relative stability with a slight decline would give, for more than two decades, the period to secondary and tertiary prevention. Finally, another window for tertiary prevention would occur in the third decade of illness, when a progression in cognitive decline could be accelerated in some cases; see Figure 2.

### 3.2. Prodromal, Onset, and Adult Life

Cognitive symptoms at the onset of puberty are considered probably the first prodromal signs of schizophrenia. Those children who developed adult schizophrenia exhibited developmental deficits in some skills that are crucial to school learning. Thus, children must cope with difficulties in verbal reasoning, working memory, attention, and processing speed as they enter primary school. Underperformance at school might, therefore, be considered one of the first signs of a latent vulnerability for schizophrenia [18].

Cognitive impairments in childhood are associated with an increased risk of schizophrenia in later life. The extent to which poor academic achievement is related to the presence of the disorder is still an open question. A meta-regression analysis, comprising data from over four million individuals, found that by the age of 16 years, those participants who later developed schizophrenia had poorer general academic and mathematics achievement than those who did not. In addition, participants with schizophrenia were less likely to enter higher education. Furthermore, participants experiencing solely psychotic-like experiences had lower general academic achievement [19].

Another important issue is that these children exhibited developmental lags, meaning their cognitive growth was slower than healthy comparison subjects. A study on a cohort of 1037 participants with a follow-up spanning 30 years where tests indexing processing speed, attention, visual–spatial problem-solving ability, and working memory showed developmental delay [20]. These findings suggest that schizophrenia may involve at least two interrelated processes: cognitive impairment and developmental lags.

Regarding cognitive development, classical theories suggest that children must move through different stages, allowing learning, maturation, and growth. Generally, cognitive development is explained as emerging from the experience-dependent development of neural structures supporting mental representations. Neural development occurs in the context of multiple interactions at different levels, from the most basic biological structures to the external environment of the developing child. Correct development depends on the integrity of these structures and the quality of interaction with the environment [21].

In the case of schizophrenia, both elements could interfere with the maturation process. On the neurobiological side, people with psychotic disorders seem to follow an abnormal neurodevelopmental trajectory, as a significant reduction in frontoparietal grey matter volume reduction has been described [22]. This reduction is significantly associated with differential age-related working memory dysfunction [23]. In addition, the grey matter deficit in children and adolescent-onset schizophrenia may occur even before the onset of the first positive symptoms [24]. During adolescence, grey matter volume in the bilateral middle temporal gyrus and reduced cortical thickness in several brain regions. These findings suggest that abnormal brain structure morphology, especially in the temporal and frontal lobes, may be an important pathophysiological feature interfering with cognitive development [25].

On the other hand, people with schizophrenia have abnormal cognitive development, as it seems most do not reach the last step of it. Usually, children go through different stages of development where they develop cognitive skills and abilities. One of the most well-established models of cognitive development is Piaget’s theory. Although the theory has been criticized, the idea of maturation evolving through different stages is still valid nowadays [26]. It proposed four stages of cognitive development: sensorimotor, preoperational, concrete operational, and formal operational. It strikes that nearly 70% of the affected people remained at the “concrete operational” stage, and only 6% of the subjects seemed to reach the last stage, called the “formal operational” [27].

The schooling period, especially between primary and secondary education, is essential for the cognitive development of humans. This development is accomplished through an interactive process between the maturational processes of the nervous system and the acquisition of cognitive skills because of interaction with the environment. The presence of cognitive impairment and delayed maturation makes people at risk of schizophrenia more vulnerable. Therefore, the schooling period is crucial to provide protective factors by implementing primary and secondary preventive strategies through remediation learning, school support, cognitive remediation programs, and other psychosocial rehabilitation interventions.

### 3.3. Later Life and Aging

Physical and cognitive decline experienced by humans starts at the beginning of adulthood and is a well-established fact [28]. Nonetheless, this process is not homogenous as it takes place differentially and at different rates for each person. Those age-related cognitive changes are caused by structural and function changes in the brain, including alterations in neuronal structure, neuronal death, loss of synapses, and dysfunction of neuronal networks [29]. Age-related diseases accelerate the rate of neuronal loss and cognitive decline, with many persons developing cognitive impairments severe enough to impair their functionality [30]. Regarding schizophrenia, some studies have shown that about 15% of older adults appear to have a substantial worsening of cognition after age 65, which leads to a general deterioration and loss of functionality.

The decline described can be particularly pronounced in people with schizophrenia. So much so that up to nearly 50% of older adults with schizophrenia may meet the criteria for dementia due to early cognitive deficits followed by age-related cognitive decline [31]. Even though this cognitive decline is like that found in neurodegenerative diseases such as Alzheimer’s, the causes may be completely different. Patterns of decline, neuropathology, and polygenetic findings, even among groups that were more chronically hospitalized, did not resemble those in Alzheimer’s disease [32,33]. Thus, minimal neurodegenerative brain pathology typical of Alzheimer’s disease pathology, such as neurofibrillary tangles or senile plaque formation, has been observed in postmortem examinations of elderly schizophrenia people [34].

In the case of age-related cognitive decline in schizophrenia, a bi-modal framework with two pathways has been proposed [35]. Pathway 1 is related to some risk factors like smoking and physical inactivity. As a result of those risk factors, metabolic syndrome would appear, and eventually, a cerebrovascular disease could be developed. In Pathway 2, the risk factors are associated with prefrontal hypoactivity. It is supposed that this low level of prefrontal activity will lead to dopamine depletion and cortical thinning as the final state. It is important to say that in both pathways, antipsychotic exposure is probably influencing the risk level of cognitive decline as it would increase dopamine depletion and augment the risk of metabolic syndrome on the other side. Moreover, levels of anticholinergic burden predicted the severity of cognitive impairment among those aged 55 years and older in regression analyses [36].

In addition, nutritional factors are closely linked to metabolic syndrome, as poor dietary habits can contribute to the development and exacerbation of metabolic-related conditions [37]. Deficiencies in essential nutrients, such as B vitamins, omega-3 fatty acids, and vitamin D, are critical for brain health, immune system, and microbiota composition and can affect cognitive function [38]. Patients with schizophrenia often exhibit irregular eating patterns, increasing their susceptibility to nutritional deficiencies. Particularly, deficits in vitamin D have been strongly associated with an elevated risk of developing schizophrenia. A systematic review and meta-analysis support this link, extending the association even to first-episode psychosis. Additionally, individuals with schizophrenia frequently experience poorer general health, unbalanced diets, lower physical activity levels, and a higher prevalence of medical comorbidities, all of which contribute to reduced circulating vitamin D levels [39]. Research has demonstrated that deficiencies in vitamin D and folate are correlated with more pronounced cognitive symptoms, even at the onset of schizophrenia [40]. Furthermore, early-life malnutrition has been proposed as a potential risk factor for schizophrenia, underscoring the critical role of proper nutrition in cognitive health [41]. Thus, addressing these deficits through diet or supplements has been proposed as an adjunctive treatment to help mitigate cognitive impairments in schizophrenia populations and enhance overall well-being [37].

Lastly, it is interesting to stress that elderly people with schizophrenia exhibit lower BDNF levels and more cognitive deficits than healthy controls, supporting the accelerated aging hypothesis of schizophrenia [42]. Accelerated aging in schizophrenia refers to the phenomenon where individuals with schizophrenia experience a faster biological aging process compared to the general population. This concept encompasses a range of physical, cognitive, and neurobiological changes that appear earlier or more rapidly in people with schizophrenia.

As it has been said, physical and cognitive functioning gradually declines from the beginning of adulthood to the later stages of the lifespan, even though the process takes place differentially and at different rates for each person. Indeed, the so-called biological age detects aspects of biological functioning that determine functional capacity more effectively than your actual chronological age [43]. Epigenetic clocks use algorithms to calculate biological age based on a readout of how well tens, or even hundreds, of sites in an individual’s genome are linked by methyl groups, a form of epigenetic modification. A recent work suggested that the analysis of the epigenetic acceleration of age is positive in schizophrenia. Additionally, long-term hospitalization rates seem to be a significant mediating factor [44].

Finally, new research at the molecular level is releasing some promising insights. For instance, the expression of G protein-coupled receptors (GPRs) has been associated with some morphological changes in brain regions responsible for cognitive impairment and behavioral changes related to schizophrenia. Different expressions of GPRs have different consequences in schizophrenia, as some increase the risk while others provide protection [45]. Future research would show whether they may be potential targets for new treatments.

### 3.4. Treating Cognition in Schizophrenia

Treating cognition in patients with schizophrenia is a clinical priority, as it is recommended in international guidelines [46]. Different strategies have been proposed, ranging from pharmacology to cognitive therapies and physical exercise. Regarding medications, no specific treatment is recommended. Nonetheless, it is advised to use second-generation antipsychotics, as they interfere less with cognition than first-generation antipsychotics. Moreover, atypical antipsychotics seem to be superior to typical at improving overall cognitive function [47]. It is important to prevent the anticholinergic and benzodiazepine burdens as they can cause impairment in cognition, and they should be kept to a minimum. Moreover, cognitive remediation and physical exercise are recommended for treating cognitive impairment in schizophrenia.

In recent years, the use of precognitive medications, such as anticholinesterases and glutamate antagonists, has gained attention in addressing cognitive symptoms associated with schizophrenia. It has long been proposed that impairments in glutamatergic signaling significantly contribute to the neuropathology of schizophrenia, especially regarding negative and cognitive symptoms [48]. Recent research has continued to explore various agents that enhance glutamatergic transmission for treating these symptoms in schizophrenia patients. However, no antipsychotics derived from the glutamatergic hypothesis have yet been approved for the treatment of schizophrenia due to mixed findings [49]. On the other hand, the cholinergic system and alpha-7 nicotinic acetylcholine receptors have also been linked to the pathophysiological mechanisms underlying cognitive impairments in schizophrenia patients [50]. Notwithstanding, to date, in the context of schizophrenia, focusing exclusively on a single pathophysiological mechanism may not yield a clinically significant outcome. Glutamatergic and cholinergic treatments, similar to other adjunctive therapies, are unlikely to demonstrate effectiveness as standalone options. Therefore, it may be necessary to combine these medications with others that possess complementary mechanisms of action [50].

Neuromodulation techniques, such as electroconvulsive therapy (ECT), deep brain stimulation (DBS), repetitive transcranial magnetic stimulation (rTMS), transcranial direct current stimulation (tDCS), transcranial random noise stimulation (tRNS), and transcranial alternating current stimulation (tACS), have been increasingly studied for their potential in improving cognitive function in schizophrenia. A recent systematic review on the cognitive effects of ECT in schizophrenia [51] found that ECT was not associated with significant cognitive deficits in patients with treatment-resistant schizophrenia (TRS). While there was some uncertainty about the immediate effects on memory, most studies showed no change or even improvement after treatment. This suggests that ECT does not have long-term negative cognitive consequences for patients with schizophrenia. Moreover, tDCS has shown promising results in improving working memory and executive functioning when applied to the dorsolateral prefrontal cortex. Transcranial direct current stimulation was associated with increased activation in the medial frontal cortex beneath the anode, showing a positive correlation with consolidated working memory performance 24 h post-stimulation [52]. DBS, a surgical procedure involving the implantation of electrodes in specific brain regions, is emerging as a potential treatment for schizophrenia. By carefully targeting areas implicated in the disorder, DBS may modulate abnormal brain circuits and alleviate symptoms. Several brain regions have been identified as potential targets for DBS in schizophrenia. The striatum, particularly the associative and ventral regions, has been extensively studied. Gault et al. [53] suggest that stimulating these areas could improve cognitive deficits and address positive and negative symptoms, respectively. Additionally, the hippocampus and cingulate cortex have shown promise as DBS targets. Perez et al. [54] found that stimulating the ventral hippocampus in animal models reduced positive symptoms and improved cognitive flexibility. Recent clinical trials have explored the efficacy of DBS for patients with treatment-resistant schizophrenia. Studies have reported positive outcomes in targeting the nucleus accumbens or subgenual anterior cingulate cortex [55]. A more recent observational study evaluated the response of DBS on clinical and cognitive outcomes in patients with schizophrenia, schizoaffective disorder, or bipolar disorder. Targeting the same brain regions, the authors reported clinical improvement without significant side effects or cognitive impairment [56].

Physical exercise interventions can be considered an evidence-based treatment to improve cognition in people living with schizophrenia [46]. Two and a half hours of moderate to vigorous physical activity per week seems to be sufficient and has been recommended to be present in multidisciplinary treatment programs [57]. In addition, meta-regression analyses indicated that greater amounts of exercise are associated with larger improvements in global cognition. Moreover, when the physical activity was supervised by professionals, improvement in cognition was more significant. Exercise significantly improved the cognitive domains of working memory, social cognition, and attention/vigilance. However, the effects on processing speed, verbal memory, visual memory and reasoning, and problem-solving were not significant [58].

Up to the moment, the strongest recommendation to treat cognition in schizophrenia is the use of cognitive remediation therapies. Different modalities of cognitive remediation are available, with pencil and paper formats, computerized programs, individual and group formats, and others. Given the heterogeneity, it is important to identify the specific elements that the treatment must have in order for it to be recognizable and proven effective. The definition most widely adopted today is the one made by the Cognitive Remediation Expert Working Group (CREW): “Cognitive remediation is an intervention that involves behavioral training aimed at cognitive problems: attention, memory, executive function, social cognition or metacognition, using the scientific principles of learning, with the ultimate goal of improving functional outcomes. Its effectiveness increases when it is provided in a context (formal or informal) that provides support and opportunity to extend to daily functioning” [59]. Therefore, the basic elements considered essential in a cognitive rehabilitation treatment are the facilitation by a therapist, the use of the massive practice of cognitive exercises, teaching skills to develop problem-solving strategies, and implementing procedures to facilitate transfer to real-world functioning [59].

Cognitive remediation has been tested in different meta-analytic analyses based on 130 studies with 8851 participants, and it can systematically produce significant benefits in cognition and functionalism in patients with schizophrenia [60,61,62,63,64]. In the different analyses, improvements with moderate effect sizes for cognition and functionalism, as well as small ones for symptoms, have been found [61,63,64]. The cognitive improvement obtained in global cognition is similar to that found in most cognitive domains [46]. Finally, the effects of treatment on global cognition appear to be long-lasting over the periods studied, between six months and two years [64].

Considering all that evidence, cognitive remediation has been considered with the highest degree of recommendation according to the guidance provided by international experts [57]. However, to be successful, certain essential elements are crucial: an active and skilled therapist, the structured development of cognitive strategies, and integration with psychosocial rehabilitation programs [63]. Therefore, based on the efficacy studies, the application of cognitive rehabilitation treatment should be a clinical recommendation of first choice in people diagnosed with schizophrenia and with cognitive impairments. It is an effective treatment with the potential to be introduced as a standard element of care rather than an optional intervention aimed at selected individuals.

The precise neurobiological processes underlying cognitive improvement in the context of cognitive remediation in schizophrenia have rarely been investigated. Some have assumed that those effects are related to neuroplasticity [65]. Neuroimaging studies suggest that cognitive remediation could facilitate neuroplasticity processes, as some changes in brain functioning have been described after cognitive therapy. Improvements in patterns of activation [66], changes in neuro-connectivity [67,68], and even some structural changes in white matter [69,70] and grey matter [71,72] have been described.

Cognitive improvement seems mediated by neurotrophic factors such as BDNF. Several studies investigated the potential link between exercise, cognitive improvement, and BDNF changes. Significant increases in cognition after exercise programs were found in some studies, along with positive correlations between BDNF and cognitive enhancements, although other studies did not observe this relationship [73]. In conclusion, the cognitive benefits of exercise in schizophrenia could be due to exercise stimulating neurogenesis, perhaps by up-regulating BDNF. Along the same line, some improvements in BDNF serum levels have been described after cognitive remediation [74,75]. Even further, epigenetic changes in the promoter zones of the BDNF gene because of cognitive remediation outcomes have also been found [76].

Another interesting and complementary approach to cognitive remediation is metacognition training. Metacognition refers to one’s ‘knowledge and cognition about cognitive phenomena’ [77]. Several different metacognitive interventions have been developed, including Metacognitive Training (MCT) [78], which seems to be the most used. It consists of various modules about psychoeducation, cognitive bias training, and strategy training and focuses on raising awareness of cognitive biases. A recent meta-analytic study [79] revealed that metacognitive therapies can improve positive symptoms, and beyond that, they have positive effects on cognitive bias and social cognition. For those reasons, measuring metacognitive abilities in schizophrenia would be critical as it would allow the monitoring of improvements or progression of the illness [80].

Thus, cognitive remediation is a possible option to ameliorate the progressive brain changes in schizophrenia to some extent, and it could help to prevent functional impairment. Physical exercise can improve cognition in individuals with schizophrenia, presumably by facilitating neuroplasticity as well.

### 3.5. Treating Cognition Across the Lifespan

Literature on cognitive remediation in the early phases of psychotic disorders is less abundant than that available for chronic disorders. Nonetheless, positive effects, ranging from mild to moderate, on cognition and functionality have been described in a meta-analysis [81]. Improvement in different domains, like verbal learning, memory, executive functions, and social cognition, has been demonstrated after cognitive interventions. Treating cognition is necessary throughout the whole lifespan. Nonetheless, it could be necessary to consider the need for personalized, stage-specific strategies to enhance cognitive function and functioning in patients (Table 1).

It is increasingly likely that a person who receives a diagnosis of a First Episode of Psychosis (FEP) had been previously diagnosed as a High-Risk Mental State (HRMS), an aspect that has been incorporated into the primary and secondary prevention of psychosis in some health services [82]. Cognitive deficits presented by people with HRMS can be described as an intermediate position between those presented by patients with a diagnosis of FEP and healthy [83].

Cognitive impairment may be more susceptible to treatment in the earliest stages of the disease, considering the potential for greater brain plasticity in these stages than in more advanced stages of the illness [84]. Consequently, the appearance of a high-risk condition for developing psychosis could be an optimal moment to apply cognitive remediation programs. Fortunately, some data cautiously support cognitive remediation interventions in adolescents and young adults in general [85]. Moreover, some evidence indicates that changes in cognition may be related to functional improvement in people with HRMS [86].

It should be noted that the different disorders that involve schizophrenia spectrum disorders, even in their earliest phases, can critically affect the ability of people who suffer from them to participate in and benefit from cognitive rehabilitation. It is essential to address motivational aspects when carrying out rehabilitation activities. Using a motivational interviewing approach, motivational reinforcement meetings, or an educational approach could be helpful in cognitive rehabilitation.

Cognitive remediation outcomes are different for young people or more aged people. It must be necessary to make some adaptations to the intervention. Corbera et al. [87] tested the hypothesis of some age-related specificity of cognitive remediation outcomes in different ages and stages of illness. They compared three groups in a randomized and controlled trial: early-stage (25 years or younger) vs. early-chronic (26–39 years old) vs. late-chronic stages (40 years old or older). Age-specific effects in improvement across three different developmental stages of schizophrenia were found. Specifically, the early-stage and early-chronic showed larger improvements in working memory, especially for those who received greater treatment duration. However, several studies have indicated that cognitive remediation still could lead to cognitive improvement in middle-aged or older patients. Wykes et al. [88] reported that patients 40 years old or older had similar improvements in their memory after cognitive remediation than those shown by younger patients. In addition, Bowie et al. [89] reported that older patients achieved significant improvements in verbal memory and verbal fluency at the same level that younger patients did. However, only the younger patients improved in other domains, such as processing speed and executive functioning.

Thus, some studies suggest that cognitive remediation can be effective also in elderly adult patients with schizophrenia. For instance, adding cognitive remediation to other psychosocial interventions in the context of comprehensive psychiatric rehabilitation led to greater improvements in logical memory and executive functioning [90]. That is an extraordinary issue as cognition is strongly related to functional competence even after considering key demographic information, such as age and education, and the anticholinergic burden of medications in older adults with schizophrenia [91]. Maybe, the key to future cognitive interventions would be to customize their procedures to target the specific cognitive characteristics of elderly people more effectively.

## 4. Conclusions

Cognitive impairment usually appears in schizophrenia before the onset of psychotic symptoms and tends to persist throughout the lifespan. Treating cognition throughout the whole lifespan is crucial for improving functional outcomes. Although there is not a particular period when treating cognition is more effective, different windows for treatment have been suggested. In general, cognitive decline began before psychosis onset, suggesting a window for primary prevention. Then, a period of relative stability with a slight decrease gives the period to secondary and, eventually, tertiary prevention for more than two decades. Finally, another window for tertiary prevention occurs from the third decade of illness until the later stages of the lifespan, when a progression in cognitive decline could be accelerated in some cases. Outcomes are different when the treatments for cognition are delivered to young people or more aged people. For that reason, some adaptations of the interventions would be necessary. Future directions in clinical research for cognitive impairment in schizophrenia should focus on precision medicine, using genetic and neuroimaging biomarkers to tailor treatments, and developing novel pro-cognitive drugs. Research on non-pharmacological approaches like cognitive remediation and metacognition-based approaches, both for therapy and cognitive assessment, is still a priority. Moreover, research on brain stimulation (e.g., TMS, tDCS) and digital interventions (e.g., remote cognitive training) will be also key. Early identification in at-risk populations and age-specific interventions will be prioritized, particularly to address cognitive decline in aging patients. Social cognition, lifestyle factors like exercise, and cross-disorder approaches are increasingly recognized as crucial for improving cognitive and functional outcomes throughout the lifespan.

## Figures and Tables

**Figure 1 healthcare-12-02196-f001:**
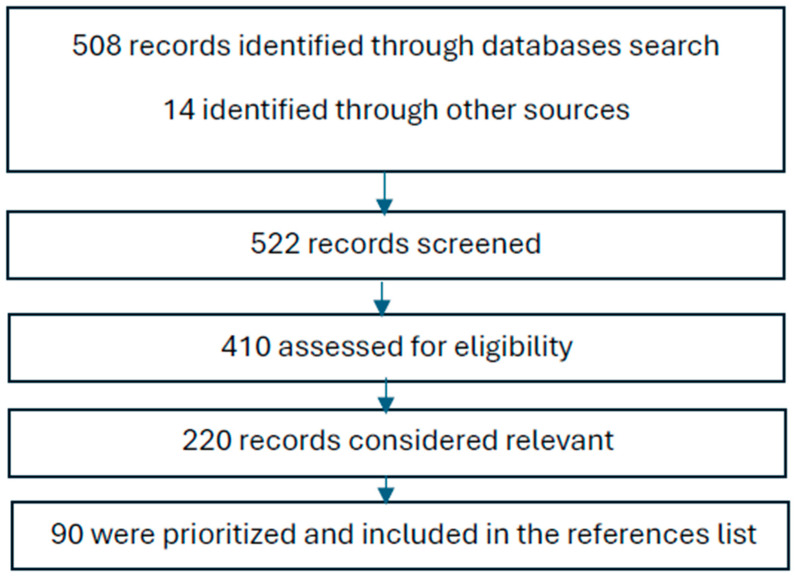
Flow chart review process.

**Figure 2 healthcare-12-02196-f002:**
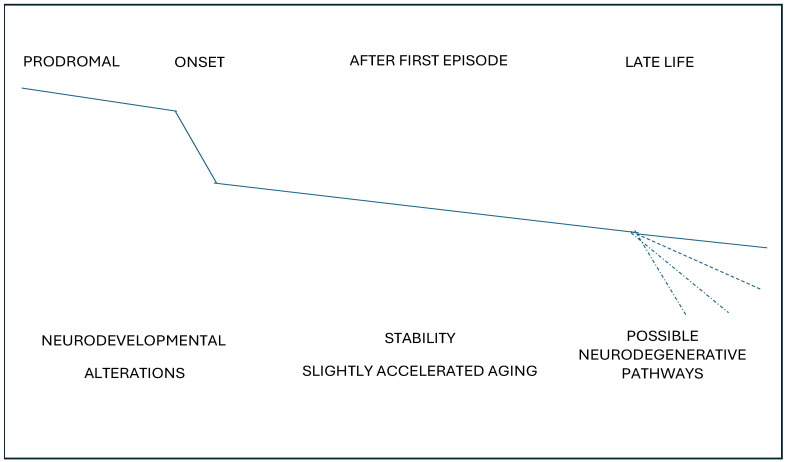
Cognitive trajectories throughout the lifespan in schizophrenia.

**Table 1 healthcare-12-02196-t001:** Treating cognition in different stages of schizophrenia.

	PRODROMAL	AFTER ONSET	LATER LIFE
** *Period* **	Before the onset	After the onset of psychosis	Third decade after the onset of psychosis
** *Cognitive interventions* **	Cognitive training Remedial learning	Cognitive remediation	Cognitive stimulation Cognitive remediation
** *Pharmacological interventions* **	Treating incipient symptoms Avoid the use of anxiolytics	Treating resistant symptoms	Treating residual symptoms Reducing anticholinergic burden
** *Other interventions* **	Physical exercise Therapeutic pedagogy	Physical exercise Psychosocial rehabilitation	Physical exercise Control vascular and other risk factors

## Data Availability

No new data were created or analyzed in this study.

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
