# Peer review of "Treating Cognition in Schizophrenia: A Whole Lifespan Perspective"

_healthcare, 2024, doi:10.3390/healthcare12212196_

Round 1
Reviewer 1 Report
Comments and Suggestions for Authors
1. This is an intesting article that discusses the cogntitive deficits in schizophrenia and their remedial measures. however, there are few suggestions that may make the article better:
a. The authors should discuss about the relevance of nutritional factors with reference to cognitive deficits in schizophrenia. As majority of the patients with schizophrenia have erratic eating habits making them vulnerable for malnutrition and deficiency of many micronutrients.
b. Role of precognitive medications like anticholine estarages and glutamate antagnoists in dealing with cognitive symptoms of schizophrenia may be discussed.
Neuromodulation techniques like ECT, rTMS, tDCS, tRNS, tACS has been used in the management of schizophrenia. Their role in cognitive enhancement may be discussed in detail.
Author Response
- This is an intesting article that discusses the cogntitive deficits in schizophrenia and their remedial measures. however, there are few suggestions that may make the article better:
- The authors should discuss about the relevance of nutritional factors with reference to cognitive deficits in schizophrenia. As majority of the patients with schizophrenia have erratic eating habits making them vulnerable for malnutrition and deficiency of many micronutrients.
We sincerely thank the reviewer for his/her meticulous review of the manuscript and the insightful suggestion. We fully agree on the importance of emphasizing the potential relevance of nutritional factors in relation to cognitive deficits in schizophrenia.
“In addition, nutritional factors are closely linked to metabolic syndrome, as poor dietary habits can contribute to the development and exacerbation of metabolic-related conditions (Zajkowska et al., 2024). Deficiencies in essential nutrients, such as B vitamins, omega-3 fatty acids, and vitamin D are critical for brain health, immune system, microbiota composition, and cognitive function (Bozzatello et al., 2024). Patients with schizophrenia often exhibit irregular eating patterns, increasing their susceptibility to nutritional deficiencies. Particularly, deficits in vitamin D have been strongly associated with an elevated risk of developing schizophrenia. A systematic review and meta-analysis support this link, extending the association even to first-episode psychosis. Additionally, individuals with schizophrenia frequently experience poorer general health, unbalanced diets, lower physical activity levels, and a higher prevalence of medical comorbidities, all of which contribute to reduced circulating vitamin D levels (Cui, 2021). Research has demonstrated that deficiencies in vitamin D and folate are correlated with more pronounced cognitive symptoms, even at the onset of schizophrenia (Firth, 2018). Furthermore, early-life malnutrition has been proposed as a potential risk factor for schizophrenia, underscoring the critical role of proper nutrition in cognitive health (Brown, 2008). Thus, addressing these deficits through diet or supplements has been proposed as an adjunctive treatment to help mitigate cognitive impairments in schizophrenia populations and enhance overall well-being (Zajkowska et al., 2024). “
- Role of precognitive medications like anticholine estarages and glutamate antagnoists in dealing with cognitive symptoms of schizophrenia may be discussed.
We fully concur on the importance of highlighting the role of precognitive medications, such as anticholinesterases and glutamate antagonists, in addressing the cognitive symptoms of schizophrenia.
“In recent years, the use of precognitive medications, such as anticholinesterases and glutamate antagonists, has gained attention in addressing cognitive symptoms associated with schizophrenia. It has long been proposed that impairments in glutamatergic signalling significantly contribute to the neuropathology of schizophrenia, especially regarding negative and cognitive symptoms (Lin et al., 2012). Recent research has continued to explore various agents that enhance glutamatergic transmission for treating these symptoms in schizophrenia patients. However, no antipsychotics derived from the glutamatergic hypothesis have yet been approved for the treatment of schizophrenia due to mixed findings (Okubo et al., 2024). On the other hand, cholinergic system and alpha-7 nicotinic acetylcholine receptors have also been linked to the pathophysiological mechanisms underlying cognitive impairments in schizophrenia patients (Koola et al., 2020). Notwithstanding, to date, in the context of schizophrenia, focusing exclusively on a single pathophysiological mechanism may not yield a clinically significant outcome. Glutamatergic and cholinergic treatments, similar to other adjunctive therapies, are unlikely to demonstrate effectiveness as standalone options. Therefore, it may be necessary to combine these medications with others that possess complementary mechanisms of action (Koola et al., 2020).”
Neuromodulation techniques like ECT, rTMS, tDCS, tRNS, tACS has been used in the management of schizophrenia. Their role in cognitive enhancement may be discussed in detail.
We fully agree that it is essential to discuss the statement regarding neuromodulation techniques in the management of schizophrenia and their role in cognitive enhancement. Consequently, we have incorporated this point into Section 3.4, ‘Treating Cognition in Schizophrenia’.
“Neuromodulation techniques, such as electroconvulsive therapy (ECT), deep brain stimulation (DBS), repetitive transcranial magnetic stimulation (rTMS), transcranial direct current stimulation (tDCS), transcranial random noise stimulation (tRNS), and transcranial alternating current stimulation (tACS), have been increasingly studied for their potential in improving cognitive function in schizophrenia.
A recent systematic review on the cognitive effects of ECT in schizophrenia (Vaccarino, 2024) found that ECT was not associated with significant cognitive deficits in patients with treatment-resistant schizophrenia (TRS). While there was some uncertainty about the immediate effects on memory, most studies showed no change or even improvement after treatment. This suggests that ECT does not have long-term negative cognitive consequences for patients with TRS. tDCS has shown promising results in improving working memory and executive functioning when applied to the dorsolateral prefrontal cortex. Transcranial direct current stimulation was associated with increased activation in the medial frontal cortex beneath the anode; showing a positive correlation with consolidated working memory performance 24 h post-stimulation (Orlov, 2017). DBS, a surgical procedure involving the implantation of electrodes in specific brain regions, is emerging as a potential treatment for schizophrenia. By carefully targeting areas implicated in the disorder, DBS may modulate abnormal brain circuits and alleviate symptoms. Several brain regions have been identified as potential targets for DBS in schizophrenia. The striatum, particularly the associative and ventral regions, has been extensively studied. Gault et al. (2018) suggest that stimulating these areas could improve cognitive deficits and address positive and negative symptoms, respectively. Additionally, the hippocampus and cingulate cortex have shown promise as DBS targets. Perez et al. (2013) found that stimulating the ventral hippocampus in animal models reduced positive symptoms and improved cognitive flexibility. Recent clinical trials have explored the efficacy of DBS for patients with treatment-resistant schizophrenia. Targeting the nucleus accumbens or subgenual anterior cingulate cortex, studies have reported positive outcomes (Corripio, 2016). A more recent observational study evaluated the response of DBS on clinical and cognitive outcomes, in patients with schizophrenia, schizoaffective disorder, or bipolar disorder. Targeting the same brain regions, the authors reported clinical improvement without significant side effects or cognitive impairment (Bioque, 2023).”

Reviewer 2 Report
Comments and Suggestions for Authors
Dear Editor, Dear Authors,
The paper under review deals with the central topic of cognitive trajectories in schizophrenia through lifespan. Though not conceived as a systematic review, the paper offers a good summary of current literature in the field and should be useful for clinicians and researchers. Nevertheless it requires some modifications to be suitable for publication.
Abstract: The abstract should include all key paper information. Please specify the number of articles screened and reviewed and the time period examined.
Materials and methods:
Though declared as a subjective, selective review, rather than systematic, more information should be added about the selection process (number of articles retriewed, screened, selected ,etc.). Also a flow diagram explaining the process shoul be useful. The process of searching, selecting or discarding papers should be explained as much as possible.
Cognitive trajectories Lane 116 - 120 Please add adequate references.
The paper could benefit from adding some evidence about the neuroanatomical and physiological basis of cognitive deficits. Also the role of modern psychopharmacological approaches should not be excluded from the dissertation, citing relevant references in the field. A review dealing with the cognitive trajectories of schizophrenia across the lifespan should not ignore the neuroanatomical and physiological basis of cognitive impairment, even with a short paragraph. Similarly, the role of psychopharmacotherapy (both in determining this impairment and in attempting to preserve cognitive function) should be recognized (De Peri et al. 2021 Psychiatry Res).
Moreover molecular mechanisms such as G-protein coupling signaling pathways have been investigated to find differences between patients with cognitive impairment and patients without negative symptoms.
Treating cognition in schizophrenia
In this context a short introduction to metacognition and its role in schizophrenia would be useful for the reader, providing appropriate references (cite for example among other relevant references Martiadis et al. 2023 Front Psychiatry; Melville G et al 2024 Psychol Med).
Conclusions should provide future directions for clinical research in cognitive impairment and treatment.
Please check the text for typos.
Comments on the Quality of English LanguagePlease check for typos.
Author Response
The paper under review deals with the central topic of cognitive trajectories in schizophrenia through lifespan. Though not conceived as a systematic review, the paper offers a good summary of current literature in the field and should be useful for clinicians and researchers. Nevertheless it requires some modifications to be suitable for publication.
Abstract: The abstract should include all key paper information. Please specify the number of articles screened and reviewed and the time period examined.
Materials and methods:
Though declared as a subjective, selective review, rather than systematic, more information should be added about the selection process (number of articles retriewed, screened, selected ,etc.). Also a flow diagram explaining the process shoul be useful. The process of searching, selecting or discarding papers should be explained as much as possible.
Thank you for your suggestions. We agree that even though this is a narrative review, adding information about the selection process improves the quality of the manuscript. We have added information about the selection process stating the number of articles retrieved, screened, and selected. Flow chart has been included.
“Publications were selected based on their representativeness of the topic, prioritizing those with the best methodological quality and greatest representativeness as it is a narrative review. The meta-analysis, systematic reviews, controlled trials, and international clinical guidelines were always preferred.”
“The literature search was conducted on May 27th, 2024, on two electronic databases (PubMed and Scopus) Abstracts were searched from 2000–2024.”
“A total of 522 publications were first identified via abstracts and titles (n =508) in databases and other sources (n = 14). “
Cognitive trajectories Lane 116 - 120 Please add adequate references.
In this paragraph there are no references as we tried to summarize and interpret the results of the previously described studies, proposing a tentative hypothesis for cognitive trajectories throughout the life cycle. To be clearer, we have rephrased the whole paragraph to emphasize those conclusions as a mere hypothesis.
“Considering the data reviewed above, we could reach the following tentative hypothesis: cognitive decline would begin before psychosis onset, suggesting a window for primary prevention. Then a period of relative stability with a slight decline would give, for more than two decades, the period to secondary and tertiary prevention. Finally, another window for tertiary prevention would occur in the third decade of illness, when a progression in cognitive decline could be accelerated in some cases.”
The paper could benefit from adding some evidence about the neuroanatomical and physiological basis of cognitive deficits. Also the role of modern psychopharmacological approaches should not be excluded from the dissertation, citing relevant references in the field. A review dealing with the cognitive trajectories of schizophrenia across the lifespan should not ignore the neuroanatomical and physiological basis of cognitive impairment, even with a short paragraph. Similarly, the role of psychopharmacotherapy (both in determining this impairment and in attempting to preserve cognitive function) should be recognized (De Peri et al. 2021 Psychiatry Res).
The reviewer is right in highlighting the importance of providing some information about the neuroanatomical basis of cognitive deficits. We have added a paragraph on that point.
“From a neuroanatomical and physiological perspective, cognitive impairment seems to be caused by different brain changes in brain structure (McCutcheon, 2023). The most reported findings are cortical thickness, ventriculomegaly, and loss of dendritic spines in pyramidal neurons of the prefrontal cortex (Martínez 2021). Those brain changes appear to have taken place in the early stages of neurodevelopment. Still, it is presumed that a cumulative effect of neurodevelopmental abnormalities is lifelong, producing enduring alterations in neuroplasticity and neuronal activity (McInstosh et al. 2011).”
Regarding pharmacotherapy:
“In recent years, the use of precognitive medications, such as anticholinesterases and glutamate antagonists, has gained attention in addressing cognitive symptoms associated with schizophrenia. It has long been proposed that impairments in glutamatergic signalling significantly contribute to the neuropathology of schizophrenia, especially regarding negative and cognitive symptoms (Lin et al., 2012). Recent research has continued to explore various agents that enhance glutamatergic transmission for treating these symptoms in schizophrenia patients. However, no antipsychotics derived from the glutamatergic hypothesis have yet been approved for the treatment of schizophrenia due to mixed findings (Okubo et al., 2024). On the other hand, cholinergic system and alpha-7 nicotinic acetylcholine receptors have also been linked to the pathophysiological mechanisms underlying cognitive impairments in schizophrenia patients (Koola et al., 2020). Notwithstanding, to date, in the context of schizophrenia, focusing exclusively on a single pathophysiological mechanism may not yield a clinically significant outcome. Glutamatergic and cholinergic treatments, similar to other adjunctive therapies, are unlikely to demonstrate effectiveness as standalone options. Therefore, it may be necessary to combine these medications with others that possess complementary mechanisms of action (Koola et al., 2020).”
Moreover molecular mechanisms such as G-protein coupling signaling pathways have been investigated to find differences between patients with cognitive impairment and patients without negative symptoms.
A brief paragraph has been included as suggested by the reviewer:
“New research at the molecular level is releasing some promising insights. For instance, the expression of G protein-coupled receptors (GPRs) has been associated with some morphological changes in brain regions responsible for cognitive impairment and behavioral changes related to schizophrenia. Different expressions of GPRs have different consequences in schizophrenia, as some increase the risk while others provide protection (Kalinovic et al 2024). Future research would show whether they may be potential targets for new treatments.”
Treating cognition in schizophrenia
In this context a short introduction to metacognition and its role in schizophrenia would be useful for the reader, providing appropriate references (cite for example among other relevant references Martiadis et al. 2023 Front Psychiatry; Melville G et al 2024 Psychol Med).
We agree with the reviewer that introducing metacognitive therapy could be of interest to the reader. Thus, we now added the following paragraph:
“Another interesting and complementary approach to cognitive remediation is metacognition training. Metacognition refers to one's ‘knowledge and cognition about cognitive phenomena’ (Flavell1979). Several different metacognitive interventions have been developed, including Metacognitive Training (MCT) (Moritz et al. 2007), which seems to be the most used. It consists of various modules about psychoeducation, cognitive bias training, and strategy training and focuses on raising awareness of cognitive biases. A recent meta-analytic study (Melville et al. 2024) revealed that metacognitive therapies can improve positive symptoms and beyond that they have positive effects on cognitive bias and social cognition. “
Conclusions should provide future directions for clinical research in cognitive impairment and treatment.
We have added a paragraph in the conclusion section:
“Future directions in clinical research for cognitive impairment in schizophrenia should focus on precision medicine, using genetic and neuroimaging biomarkers to tailor treatments, and developing novel pro-cognitive drugs. Research on non-pharmacological approaches like cognitive remediation therapy, brain stimulation (e.g., TMS, tDCS), and digital interventions (e.g., remote cognitive training) will be also key. Early identification in at-risk populations and age-specific interventions will be prioritized, particularly to address cognitive decline in aging patients. Social cognition, lifestyle factors like exercise, and cross-disorder approaches are increasingly recognized as crucial for improving cognitive and functional outcomes throughout the lifespan.”
Please check the text for typos.
We have checked and corrected the text for typos.

Round 2
Reviewer 2 Report
Comments and Suggestions for Authors
Dear Authors,
Thank you for your careful review according to our instructions.
In my opinion, some final changes are still needed. Lane 382 - 389: "Another interesting and complementary approach to cognitive remediation is metacognition training. Metacognition refers to one's ‘knowledge and cognition about cognitive phenomena’ [77]. Several different metacognitive interventions have been developed, including Metacognitive Training (MCT) [78], which seems to be the most used. It consists of various modules about psychoeducation, cognitive bias training, and strategy training and focuses on raising awareness of cognitive biases. A recent meta-analytic study [79] revealed that metacognitive therapies can improve positive symptoms and beyond that they have positive effects on cognitive bias and social cognition."
A brief mention of the possibility of accurately measuring metacognitive abilities in schizophrenia, allowing the monitoring of improvements or progression of the illness, should be appreciated (citing the recent review by Martiadis et al. 2023 Frontiers in Psychiatry, which is the most up-to-date collection of metacognition assessment tools in schizophrenia).
Please consider including in future research directions (conclusions) some lines on the need for further research in cognitive remediation and metacognition-based approaches, both for therapy and cognitive assessment.
Thanks a lot for your work.
Author Response
In my opinion, some final changes are still needed. Lane 382 - 389:
"Another interesting and complementary approach to cognitive remediation is metacognition training. Metacognition refers to one's ‘knowledge and cognition about cognitive phenomena’ [77]. Several different metacognitive interventions have been developed, including Metacognitive Training (MCT) [78], which seems to be the most used. It consists of various modules about psychoeducation, cognitive bias training, and strategy training and focuses on raising awareness of cognitive biases. A recent meta-analytic study [79] revealed that metacognitive therapies can improve positive symptoms and beyond that they have positive effects on cognitive bias and social cognition."
A brief mention of the possibility of accurately measuring metacognitive abilities in schizophrenia, allowing the monitoring of improvements or progression of the illness, should be appreciated (citing the recent review by Martiadis et al. 2023 Frontiers in Psychiatry, which is the most up-to-date collection of metacognition assessment tools in schizophrenia).
Response: Thank you for the suggestion. We have now mentioned the importance of measuring metacognitive abilities and added the suggested reference.
“For those reasons, measuring metacognitive abilities in schizophrenia would be critical as it would allow the monitoring of improvements or progression of the illness (Martiadis et al. 2023).”
Please consider including in future research directions (conclusions) some lines on the need for further research in cognitive remediation and metacognition-based approaches, both for therapy and cognitive assessment.
Response: We have added the suggestion about cognitive remediation and metacognition-based approaches in conclusions as suggested.
“Research on non-pharmacological approaches like cognitive remediation and metacognition-based approaches, both for therapy and cognitive assessment is still a priority.”
Thanks a lot for the review.
